# MbWRKY53, a *M. baccata* WRKY Transcription Factor, Contributes to Cold and Drought Stress Tolerance in Transgenic *Arabidopsis thaliana*

**DOI:** 10.3390/ijms25147626

**Published:** 2024-07-11

**Authors:** Wanda Liu, Tianhe Wang, Xiaoqi Liang, Qinglei Ye, Yu Wang, Jilong Han, Deguo Han

**Affiliations:** 1Horticulture Branch, Heilongjiang Academy of Agricultural Sciences, Harbin 150069, Chinahaaswth@126.com (T.W.); haaswangyu@126.com (Y.W.); hanjilong0000@163.com (J.H.); 2Key Laboratory of Biology and Genetic Improvement of Horticultural Crops (Northeast Region), Ministry of Agriculture and Rural Affairs, National-Local Joint Engineering Research Center for Development and Utilization of Small Fruits in Cold Regions, College of Horticulture & Landscape Architecture, Northeast Agricultural University, Harbin 150038, China; a13980332297@163.com; 3Heilongjiang Agricultural Technology Extension Station, Harbin 150090, China; yeqinglei2008@163.com

**Keywords:** *Malus baccata* (L.) Borkh, WRKY transcription factor, low temperature stress, drought stress

## Abstract

Apple is an important horticultural crop, but various adverse environmental factors can threaten the quality and yield of its fruits. The ability of apples to resist stress mainly depends on the rootstock. *Malus baccata* (L.) Borkh. is a commonly used rootstock in Northeast China. In this study, it was used as the experimental material, and the target gene *MbWRKY53* was screened through transcriptome analysis and Real-Time Quantitative Reverse Transcription Polymerase Chain Reaction (RT-qPCR) after cold and drought treatment. Bioinformatics analysis revealed that this transcription factor (TF) belonged to the WRKY TF family, and its encoded protein was localized in the nucleus. RT-qPCR showed that the gene was more easily expressed in roots and young leaves and is more responsive to cold and drought stimuli. Functional validation in *Arabidopsis thaliana* confirmed that *MbWRKY53* can enhance plant tolerance to cold and drought stress. Furthermore, by analyzing the expression levels of genes related to cold and drought stress in transgenic *Arabidopsis* lines, it was inferred that this gene can regulate the expression of stress-related genes through multiple pathways such as the CBF pathway, SOS pathway, Pro synthesis pathway, and ABA-dependent pathways, enhancing the adaptability of transgenic *Arabidopsis* to cold and drought environments.

## 1. Introduction

Plants are sessile organisms, but their surrounding environment is constantly changing. As a result, plants are not only threatened by biological factors such as bacteria, fungi, weeds, pests, and parasitic plants [1] but also face various abiotic threats, including salinity, alkalinity, flooding, high temperatures, low temperatures, and drought [2]. These adverse conditions limit the normal life activities of plants. However, plants have evolved mechanisms to respond to these stresses. When plants perceive these adverse environmental factors, they can transmit them to various organs and tissues and immediately initiate corresponding responses. For example, the expression levels of relevant resistance genes change, the synthesis rate of antioxidant substances increases, and cell wall stability is enhanced, among other adaptive changes, in order to minimize the damage caused by stress to plants [3,4,5]. When the environmental threat subsides, these response mechanisms can return to their pre-stress state, continuing to maintain various life activities of the plant. In these response mechanisms, transcription factors (TFs) play an important role. TFs are a class of protein molecules that can bind to specific sequences in DNA, and they can either promote or inhibit gene transcription [6].

WRKY TF is one of the most widely distributed classes in plants, named for its conserved WRKY domain. The first WRKY protein was cloned from sweet potato (*Ipomoea batatas*) in 1994 [7], and since then, a large number of WRKY TFs have been identified and studied in various plants, such as *A. thaliana* [8], rice [9], rapeseed [10], and some lower plants [11]. Research has shown that WRKY TFs play important roles in regulating plant growth and development, stress responses, light signal transduction, hormone regulation, and many other biological processes by binding to cis-acting elements in the target genes. For example, it has been demonstrated that the WRKY45 protein in rice (*Oryza sativa*) can affect root growth and differentiation by regulating the expression of genes related to auxin synthesis and transport. *PheWRKY86* can increase its expression under drought and ABA induction and further enhance plant tolerance by regulating the expression of *NCED1* [12]. Under cold stress, *CsWRKY46* in cucumber, stimulated by cold factors, promotes the expression of downstream stress genes *RD29A* and *COR47* through the ABA pathway, conferring enhanced cold resistance to transgenic plants [13]. The WRKY33 TF interacted with the key enzyme SnRK2.4 kinase in the ABA signaling pathway, not only enhancing the transmission of ABA signals but also increasing the plant’s survival ability under drought conditions [14].

Apples (*Malus* × *domestica*) are considered one of the top four fruits in the world, being rich in various nutrients and having high economic value. Its cultivation range is spread all over the world; however, due to the impact of abiotic stress, the development of the apple industry is greatly restricted [15]. Particularly in the northeastern regions of China, where temperatures are low and cold periods are long, posing a major threat to the apple industry. If exposed to low temperatures for a long time, the morphology, growth, and metabolic pathways of fruit trees will be severely affected, leading to a reduced or even non-existent harvest [16]. Additionally, drought is another major challenge faced by apple cultivation in China [17]. Therefore, the urgent task of breeding new apple varieties with cold and drought tolerance is evident. In this study, by combining previous research and studying the functions of TFs in apples, as well as exploring potential genes related to cold and drought resistance, we can provide a theoretical basis for the study of apple stress resistance and the cultivation of new varieties.

## 2. Results

### 2.1. The Conserved Structure and Physicochemical Properties of the MbWRKY53 Protein

After analyzing the predicted protein MbWRKY53 on the ProtParam website (http://www.expasy.org/tools/protparam.html, accessed on 20 November 2023), it was found that the protein was composed of 351 amino acids (aa), containing 20 types of aa. The most abundant amino acids were Ser (12.8%), Leu (7.1%), and Pro (6.8%). The predicted molecular weight was 39.40 kDa, with a theoretical pI of 5.61. The average hydrophilicity was −0.807, indicating strong hydrophilicity. The instability index was 57.55, suggesting that it was an unstable protein. The open reading frame (ORF) of MbWRKY53 was 1056 bp and contained a WRKYGQK conserved domain specific to WRKY TFs, indicating that it belonged to the WRKY TF family (Appendix A).

### 2.2. The Homology Analysis of the MbWRKY53 Protein

Through alignment analysis of the highly homologous sequences of MbWRKY53 with 10 sequences from different species, it was found that these sequences exhibit high similarity and the same ‘WRKY’ motif (Figure 1A). The sequences included PbWRKY53 (*Pyrus bretschneideri*, XP_048422667.1), PmWRKY53 (*Prunus mume*, XP_008239051.1), CfWRKY53 (*Cornus florida*, XP_059642829.1), MnWRKY53 (*Morus notabilis*, XP_010104968.1), QrWRKY53 (*Quercus robur*, XP_050269032.1), RvWRKY53 (*Rhododendron vialii*, XP_058223914.1), AaWRKY53 (*Argentina anserina*, XP_050378931.1), RrWRKY53 (*Rosa rugosa*, XP_062027121.1), RcWRKY53 (*Rosa chinensis*, XP_024170896.1), and AtWRKY53 (*Arabidopsis thaliana*, CAD5328853.1). A phylogenetic tree was constructed for these similar proteins, revealing that the PbWRKY53 was most closely related to MbWRKY53 (Figure 1B). These results provided important references for further research on the function and evolution of the MbWRKY53 protein.

### 2.3. Structural Prediction of MbWRKY53 Protein

It was found through SMART program analysis that there is one WRKYGQK domain and three low-complexity regions in the protein encoded by MbWRKY53 (Figure 2B). Then, its tertiary structure was predicted through the SWISS-MODEL website, and the results showed that the tertiary structure of this protein was consistent with its secondary structure. The secondary structure contains 23.36% α helix, 9.69% extended strand, 3.70% β turn, and 63.25% disordered coil (Figure 2A). These results provide important clues for further exploration of the function and structure of the MbWRKY53 protein.

### 2.4. Subcellular Localization of MbWRKY53 Protein

The fluorescent distribution of MbWRKY53-GFP under confocal microscopy is shown in Figure 3. In Figure 3B, it can be observed that the green fluorescence of GFP was distributed throughout the entire cytoplasm, while in Figure 3E, only the green fluorescence of the MbWRKY53-GFP fusion protein was observed in the cell nucleus. The position of the nucleus was confirmed by staining with 4′,6-diamidino-2-phenylindole (DAPI), as shown in Figure 3F. Therefore, it can be demonstrated that the MbWRKY53 protein was a nuclear protein.

### 2.5. The Spatiotemporal Expression of MbWRKY53

The expression levels of the target gene in different tissues of *M. baccata* (root, stem, young leaves, and mature leaves) were analyzed by Real-time fluorescence quantification (RT-qPCR), and it was found that its expression levels varied in these tissues, with *MbWRKY53* showing higher expression in young leaves and roots. Therefore, the young leaves and roots were selected as the focus of the next step of the experiment. Samples from these tissues were collected after exposure to low temperatures, high temperatures, drought, ABA, and salt treatments; total RNA was extracted, reverse transcribed, and analyzed by RT-qPCR. The results indicated that the expression levels of MbWRKY53 in both tissues showed a similar trend, increasing first and then decreasing over time (Figure 4). Additionally, it was observed that this gene was more sensitive to low temperature and drought stress because the peak expression levels of *MbWRKY53* were higher under these two stress conditions, and the time to reach the peak was shorter. Taking the expression level of the gene under untreated conditions as the control, under low-temperature stress, the expression levels of the gene reached the peak at 4 h in both roots and young leaves, and they were 11.70-fold and 8.33-fold that of the control, respectively; under drought stress, its expression level reached the peak at 6 h in roots, which was 12.39-fold that of the control, while in young leaves, the peak was reached at 4 h, which was 10.52-fold that of the control. Based on these results, further research would be conducted to elucidate the underlying mechanisms of *MbWRKY53* in plant responses to low-temperature and drought stress.

### 2.6. Analysis of Cold and Drought Tolerance of MbWRKY53-OE Arabidopsis

After RT-qPCR analysis of the expression levels of MbWRKY53 in transgenic *Arabidopsis*, the expression levels of *MbWRKY53* in the six transgenic lines were 1.63, 2.41, 2.15, 1.93, 1.72, and 1, respectively. The three lines with the highest expression levels, namely L2, L3, and L4, were selected for the next step of the study (Appendix A). After growing under low-temperature conditions (4 °C), various degrees of browning and wilting were observed in the leaves, particularly in the wild-type (WT) and vector control (UL) lines, where the changes were more pronounced, as shown in Figure 5A. However, after being returned to a normal growth environment for 7 d, the overexpressing lines gradually regained their normal phenotype, with the majority of plants maintaining good growth status, resulting in survival rates of 84.21% (L2), 80.2% (L3), and 81.32% (L4). In contrast, the survival rates of the WT and UL lines were only 17.65% and 18.55%, respectively. It was noteworthy that before treatment, the survival rates of all lines were generally above 90% (Figure 5C).

Similarly, after 10 days of water shortage treatment, the WT and UL lines displayed severe wilting and yellowing, while the phenotype of lines L2/3/4 did not show significant changes (Figure 5B). Following rehydration at the end of the treatment, one week later, the survival rates of the WT and UL lines were only 20.11% and 19.24%, respectively, whereas the overexpressing lines gradually recovered to normal levels with survival rates of 83.67% (L2), 84.33% (L3), and 85.21% (L4) (Figure 5C). These results indicated that overexpression of *MbWRKY53* had a positive regulatory effect on plant responses to low temperature and drought stress.

To further analyze the impact of *MbWRKY53* overexpression on plant cold and drought tolerance, changes in substances such as the content of chlorophyll (chl), malondialdehyde (MDA), and proline (Pro), as well as the activities of catalase (CAT), superoxide dismutase (SOD), and peroxidase (POD) before and after low temperature and drought treatments were measured and compared. These physiological and biochemical indicators can be used to evaluate the degree to which plants are affected by stress. Figure 6 shows the comparison results, and before treatment, these indicators showed little difference among all *Arabidopsis* lines. However, after stress treatment, there were significant differences in these indicators among different lines. Chl content decreased in all lines post-treatment, but the reduction was less pronounced in the overexpressing lines and significantly higher than in the WT and UL lines. Conversely, there was an increase in MDA content and proline content, as well as the activities of SOD, POD, and CAT in all lines post-treatment. However, MDA content was higher in the WT and UL lines compared to the overexpressing lines, while the activities of SOD, POD, and CAT were lower in the WT and UL lines than in the overexpressing lines. These data demonstrated that the overexpression of *MbWRKY53* in *Arabidopsis* effectively enhanced plant cold and drought tolerance and improved antioxidant capacity.

### 2.7. MbWRKY53 Induces the Expression of Cold and Drought-Resistant Genes in Arabidopsis

In order to elucidate the regulatory mechanism of *MbWRKY53* in response to stress, leaf samples of *Arabidopsis* were collected and subjected to RT-qPCR analysis to measure the expression levels of target genes related to low temperatures and drought stress, including *SOS1*, *COR47*, *DREB2A*, *P5CS1*, *COR6.6*, and *RD29b*. The gene expression changes are shown in Figure 7, indicating that under normal conditions, the expression levels of these genes remained relatively low in all plants. However, after stress treatment, the expression levels of these six target genes increased, with varying degrees of change. The expression levels of these genes in the overexpressing lines were much higher than in the WT and UL lines. Therefore, it can be concluded that *MbWRKY53* regulated plant responses to low temperatures and drought stress by modulating the expression of these target genes. This suggested that *MbWRKY53* played a significant regulatory role in plant stress responses.

## 3. Discussion

WRKY TF, as a large family in plants, plays crucial roles in plant growth, development, and stress resistance. A notable characteristic of this family is the presence of a conserved domain containing a unique structure formed by a disulfide bond between two cysteine residues [18,19]. WRKY TFs bind to specific DNA sequences (W-box or its variants) of target genes, thereby affecting the expression of these specific genes to regulate plant growth and development and response to environmental stress [20]. Furthermore, WRKY TFs can form complexes with other proteins to further modulate gene expression. In fruit trees, the expression of *WRKY* genes is closely associated with fruit sugar accumulation, antioxidant systems, and fruit ripening [21]. When plants are subjected to diseases or environmental stress, the expression of WRKY genes also changes. For example, the expression level of wheat *TaWRKY1-2D* varies under drought stress, highlighting its importance as a candidate gene for drought resistance in plants [22]. Additionally, WRKY TFs also play key roles in studies of disease resistance and stress tolerance in pears and peaches. These research findings indicate that WRKY TFs have broad prospects for application in fruit tree biology and plant pathology.

In this study, the sequence characteristics of the target gene were analyzed, and the results showed that the ORF length of *MbWRKY53* was 573 bp, encoding 190 aa, and the predicted molecular weight of the encoded protein was 21.70 kDa. The MbWRKY53 protein belonged to an unstable hydrophilic protein. In addition, it contained the WRKYGQK conserved sequence unique to the WRKY family and the C2H2 zinc finger domain, a significant feature indicating that this TF belongs to the WRKY family [23]. These findings further enrich the membership of the family. Evolutionary tree analysis showed that the genetic relationship between MbWRKY53 and PbWRKY53 was closest. After structural analysis of the target protein, it was found that the secondary and tertiary structures of MbWRKY53 were similar to other WRKY TFs, further indicated that it belongs to this family and can regulate plant response to stress by binding to W-boxes or other functional elements of target genes [24]. Subcellular localization showed that this protein was located in the cell nucleus, suggesting that it may play a transcriptional regulatory role in the nucleus.

When studying the function of the target gene, it is crucial to determine its expression levels in different plant tissues. Comparing the expression patterns in different tissues can reveal the specific functions of genes within cells and tissues, which may be related to the adaptability and stress resistance of plants to the environment. Previous studies have demonstrated that gene expression is tissue-specific, with varying expression levels in different organs [25]. For example, in kiwifruit, *AchLOX3*, *AchOPR3*, and *AchABF1-1* exhibit inconsistent expression levels in different tissues, among which *AchLOX3* is more easily expressed in roots, buds, and leaves; *AchOPR3* was highly expressed in leaves; and *AchABF1-1* had higher expression levels in roots, shoots, leaves, and fruits [26]. In this study, we analyzed the expression levels of *MbWRKY53* in the roots, stems, and leaves (both young and mature) of *M. baccata*. It was found that the gene was more easily expressed in young leaves and roots; therefore, we chose these two tissues for further research. After undergoing different stress treatments on *M. baccata*, we discovered that *MbWRKY53* was more responsive to low temperatures and drought stresses. Based on these results, we tentatively hypothesize that *MbWRKY53* might be a crucial candidate gene for plant resistance to cold and drought. To validate this hypothesis, further research was conducted to explore the specific functions and regulatory mechanisms of this gene.

To verify its function, the gene was transferred into the model plant *Arabidopsis* and analyzed from phenotypic, physiological, and biochemical aspects. The results indicated there were noticeable changes in the phenotypes of the WT, UL, and *MbWRKY53*-overexpressing (*MbWRKY53*-OE) *Arabidopsis* plants, but the degree of change varied significantly. Under low-temperature treatment, the leaves of WT and UL lines showed significant frost damage and browning, while the damage to transgenic lines was relatively mild. Under drought treatment, the leaves and stems of WT and UL lines showed obvious wilting and a large number of deaths, while transgenic lines showed no significant changes and a higher survival rate. These changes indicated that gene overexpressing lines exhibited higher cold and drought tolerance after low temperature and drought treatments. In order to further confirm the function of the transgenic *MbWRKY53* gene in improving plant cold and drought resistance, physiological indicators were measured after cold and water stress.

Environmental stress can inhibit plant photosynthesis, leading to disruption of chl synthesis and stability, resulting in a decrease in chl content [27]. The results showed that in this study, chl content showed a decreasing trend in all lines, but the decrease in chl content was less in overexpressing lines, and it was significantly higher than WT and UL lines. MDA is a marker of cell membrane damage, and an increase in its content can reflect the degree of oxidative damage caused by stress [28]. After stress treatment, the MDA content in transgenic plants was significantly lower than that in the control group. Proline (Pro), as an important antioxidant and osmoregulatory substance, plays an important role in protecting cell membrane integrity and regulating cell osmotic regulation; *P5CS1* can regulate plant response to abiotic stress by synthesizing proline [29]. In addition, stress can also lead to the accumulation of ROS in plants. High levels of ROS can cause oxidative damage to biological molecules such as cell membranes, proteins, and nucleic acids, thereby affecting plant growth and development [30]. To counteract these harmful ROS, plants regulate the activities of antioxidant enzymes such as SOD, POD, and CAT to maintain redox balance, thereby enhancing plant adaptation to cold and drought [31,32]. After stress, the WT and UL lines showed higher increases in MDA content, H_2_O_2_ content, O_2_^−^ content, and relative conductivity, while the activities of SOD, POD, and CAT were lower. This indicated that the *MbWRKY53*-OE lines had a stronger capacity to scavenge ROS, thereby alleviating the oxidative damage caused by cold and drought stress. These results further confirm that *MbWRKY53* effectively enhances plant cold and drought tolerance.

WRKY TF can regulate the transcription of target genes by directly binding to the promoter elements of these genes, which plays an important role in regulating plant response to stress [33]. In order to investigate the regulatory mechanism of *MbWRKY53* on downstream genes, RT-qPCR was used to detect expression levels of target genes related to the CBF pathway, SOS pathway, ABA-dependent pathway, and Pro synthesis pathway under cold and drought stress, including *SOS1*, *COR47*, *DREB2A*, *P5CS1*, *COR6.6*, and *RD29b*. It was found that under non-stress conditions, the expression levels of these genes were maintained at relatively low levels in all plants. However, after stress treatment, the expression levels of these six target genes all increased, although the degree of change varied. Importantly, in the overexpressing plant lines, the expression levels of these genes were significantly higher than those in the control group (WT, UL).

Based on previous studies and the results of this study, we have described a possible pathway by which *MbWRKY53* regulates plant response to low temperatures and drought stress (Figure 8). After receiving low-temperature and drought stimuli, the expression level of *MbWRKY53* increases. Then, by binding to cis-acting elements of CBF or participating in ABA synthesis and SOS pathways, it activated and upregulated the expression of these related genes, improving the survival ability of plants in low-temperature and drought environments.

## 4. Materials and Methods

### 4.1. Cultivation and Treatment of M. baccata

First, the *M. baccata* seeds were subjected to germination treatment (4 °C, 10 d), followed by placement under warm conditions. After germination, the seeds were sown in a nutrient soil composed of soil and vermiculite at a ratio of 2:1. Subsequently, the sown seeds were placed in a cultivation chamber at 25 °C and watered regularly [34]. After two weeks, seedlings with good growth rates were selected and placed in Hoagland nutrient solution for hydroponic treatment. The nutrient solution replacement cycle is 3 d [35]. After the seedlings developed roots and approximately 8 fully expanded leaves, 30 well-growing seedlings were selected and divided into six groups for different treatments: (1) Control group: placed in Hoagland nutrient solution at room temperature; (2) Low-temperature treatment: cultivated at 4 °C; (3) High-temperature treatment: cultivated at 37 °C; (4) Drought treatment: placed in 20% PEG6000 (Solarbio, P8250, Beijing, China) nutrient solution at room temperature; (5) High-salt treatment: placed in 200 mM NaCl (Solarbio, LA0200, Beijing, China), hydroponic solution contained at room temperature; (6) ABA (Solarbio, A8060, Beijing, China) treatment: placed in hydroponic solution containing 50 μM ABA at room temperature. Within the first 12 h of treatment, samples of the roots, stems, and leaves of the seedlings were collected every two hours, immediately frozen in liquid nitrogen, and stored at −80 °C.

### 4.2. Sowing and Cultivation of Arabidopsis

Colombian WT *Arabidopsis* seeds, culture media, and culture dishes were prepared, and sterile operations were prepared on an ultra-clean workbench. *Arabidopsis* seeds were placed in a 1.5 mL centrifuge tube, added 70% anhydrous ethanol was used for initial disinfection; then, the seeds were cleaned with sterile water, soaked in a 5% sodium hypochlorite solution for further disinfection, and, finally, the seeds were repeatedly cleaned with sterile water. Spread the cleaned and disinfected seeds evenly on sterile filter paper to dry. Using sterilized toothpicks, the seeds were spread evenly on the surface of the culture medium to avoid sowing too densely. The culture dish was sealed after sowing, then refrigerated at 4 °C for 2–3 d to allow the seeds to germinate [36]. After germination, they were transferred to a nutrient bowl (peat soil: vermiculite = 2:1), and four seedlings were planted in each bowl.

### 4.3. Cloning and Sequencing of the MbWRKY53 Gene

The total RNA of the samples was stored in 2.1.1 using the OminiPlant RNA Kit (CW2598S, 50preps) from Kangwei Century Company (Beijng, China). We used the reverse transcription kit (product number: AE311-02) of TransGen Biotech (Beijng, China) for reverse transcription, detected the purified cDNA by agarose gel electrophoresis, and ensured its integrity and purity [37].

Based on the CDs region of the *MdWRKY53* (MD06G1104100) nucleic acid sequence of apples found in NCBI (https://www.ncbi.nlm.nih.gov/, accessed on 25 March 2023), two pairs of specific primers were designed in Primer 5.0 software to amplify the target gene. The primer sequences are displayed in Appendix A. The PCR reaction system is shown in Appendix A, and the reaction conditions is shown in Appendix A. A total of 35 cycles were conducted.

After ligating the synthesized PCR products with the *pEASY^®^*-T5 cloning vector (TransGen, Beijing, China, CT501), we transformed the vector into Trans1-T1 competent cells (CD501). Subsequently, the transformed cells were screened, positive clones were selected for validation and sent to BGI (Beijing) for sequencing [38].

### 4.4. Bioinformatic Analysis of the MbWRKY53 Gene

The sequencing results of the MbWRKY53 gene were aligned using EMBOSS Needle (http://www.ebi.ac.uk/Tools/psa/emboss_needle/, accessed on 20 May 2023), followed by translation of the nucleotide sequence into aa using DNAMAN 5.2 software. By utilizing the BLAST function of NCBI, similarity alignment was performed on the translated aa sequences, and a homologous evolutionary tree was constructed using Mega7.0 to evaluate their phylogenetic relationships, including aa sequences from 10 other homologous evolutionary branches in the evolutionary tree. Predicted the primary structure, secondary structure, domain, and tertiary structure of the target protein using the ExPASy platform (https://web.expasy.org/protparam/, accessed on 10 May 2023), SOPMA (https://npsa-prabi.ibcp.fr/, accessed on 10 March 2023), SMART tool (https://swissmodel.expasy.org/, accessed on 20 May 2023), and SWISS-MODEL platform (http://smart.embl-heidelberg.de/, accessed on 20 May 2023) for prediction, respectively.

### 4.5. Subcellular Localization of MbWRKY53 Protein

Using CaMV 35S-sGFP pCAMBIA1300 as the expression vector, *BamH*I and *Kpn*I were selected as restriction endonuclease sites on the vector, and upstream and downstream primers (*MbWRKY53*-2F/R) containing these two sites were designed (Appendix A). Subsequently, the target protein MbWRKY53 and the vector were subjected to double enzyme digestion, and the target fragment of MbWRKY53 was inserted into the plasmid vector to construct a transient expression vector for MbWRKY53. Gold powder containing MbWRKY53 plasmid and control plasmid was injected into onion epidermal cells using the particle bombardment method [39]. Finally, the localization of the target protein was observed using a confocal microscope (LSM 510 Meta, Zeiss, Oberkochen, Germany). The excitation light has a wavelength of 587 nm, and the emission light receives a wavelength range of 607–650 nm.

### 4.6. Analysis of the Expression Level of the MbWRKY53 Gene

Design specific primers *MbWRKY53*-qF/R based on the conserved domain of *MbWRKY53*, with primer sequences shown in Appendix A. The cDNA obtained in Section 2.2 was used as the template for the reaction, and *Mbactin* was selected as the internal reference gene [40]. According to the standard response systems and procedures in Appendix A, RT-qPCR detection was performed on the expression levels of *MbWRKY53* in the roots, stems, young leaves, and mature leaves of *M. baccata* under different stress conditions. Subsequently, the relative transcription levels were analyzed using the 2^−ΔΔΔCT^ method [41].

### 4.7. Obtaining Genetically Modified Arabidopsis

The pCAMBIA2300 plasmid was selected as the vector, and dual restriction endonucleases with *BamH*I and *Kpn*I were performed to obtain a linearized vector. By connecting the *MbWRKY53* gene fragment with a linearization vector to obtain the overexpression vector pCAMBIA2300-*MbWRKY53*. The vector was transformed into Agrobacterium tumefaciens GV3101 competent cells (AC1001), and *Agrobacterium tumefaciens* was transferred to Colombian *Arabidopsis* by using inflorescence-mediated methods [42]. The seeds were collected from different lines after maturity, and the infected seeds were marked as T_1_, while the uninfected seeds were marked as T_0_. Transferring T_1_ generation seeds onto MS screening medium containing 50 mg/L kanamycin, it can be preliminarily considered that *Arabidopsis* had been successfully transformed and can germinate normally. The transgenic seeds were collected for further screening to obtain T_2_ generation plants, and, finally, transgenic *Arabidopsis* was determined through RT-qPCR analysis. The three lines with the highest expression levels were selected for cultivation to obtain T_3_-generation transgenic plants. WT type and UL type were used as control groups, and the next step of treatment was carried out on these three lines [43].

### 4.8. Determination of Relevant Physiological Indicators

The *Arabidopsis* was transplanted into nutrient soil and grew for about 20 d. The control group and transgenic type were divided into 2 groups, with 10 seedlings per plant line in each group. The first group was subjected to low-temperature stress treatment by being placed in a −4 °C incubator for 10 h. The second group was subjected to drought stress treatment: under sufficient water conditions, watering was stopped for 10 d. After the stress was over, the plants were allowed to grow under normal conditions for 7 d, the survival rate of each plant line was calculated, and the phenotypic changes of the plants were recorded [44]. In addition, the relevant physiological indicators of all untreated and stressed strains were measured, including chl content using the ethanol extraction method [45], SOD activity using the nitrogen blue tetrazole photoreduction method [46], POD activity using the guaiacol method [46], CAT activity using UV absorption method [46], MDA content using spectrophotometer colorimetric method [47], Pro content using sulfosalicylic acid extraction method [48], H_2_O_2_ content using titanium reagent method [49], O_2_^−^ content using hydroxylamine oxidation method [49], and relative conductivity using extraction method.

### 4.9. Expression Analysis of Stress-Related Key Genes in Arabidopsis

The RNA of WT, UL, and the overexpressed Arabidopsis plants was extracted before and after stress treatment, and template cDNA was obtained through reverse transcription. The purity of RNA can be evaluated by agarose gel electrophoresis. High-quality RNA would show clear 28S, 18S, and 5S rRNA bands, and the brightness of the 28S band is approximately twice that of the 18S band. Genomic DNA residues were removed using the gDNA Removal Kit (AmyJet Scientific Inc., Wuhan, China). Then, specific primers (Appendix A) were used to analyze the genes related to low-temperature and drought stress in *Arabidopsis* by RT-qPCR with Thermo Fisher QuantStudio 5, (Thermo Scientific, New York, NY, USA), and the qPCR results were analyzed by referring to the method of Livak and Schmittgen [50]. The reaction system and reaction procedures of RT-qPCR are shown in Appendix A.

### 4.10. Data Analysis

We performed one-way ANOVA by using SPSS 21.0 software (IBM, Chicago, IL, USA) on the mean values of all indicators obtained from three repeated tests, and the standard deviation (SD) of the data was measured to evaluate the mean differences between groups [51]. The calculated significant difference (*p*-value) can be used to reflect the difference in experimental results. When *p* ≤ 0.05, it indicated a significant difference between the two sets of data (* *p* ≤ 0.05), and when *p* ≤ 0.01, it indicated a very significant difference between the two sets of data (** *p* ≤ 0.01) [52].

## 5. Conclusions

In this study, we successfully cloned and isolated *MbWRKY53* from *M. baccata*, which belonged to the WRKY transcription factor gene. The MbWRKY53 protein was located in the nucleus and was a hydrophilic unstable protein with the closest genetic relationship to PbWRKY53. *MbWRKY53* can be induced expression under various stress conditions, such as low-temperature, high-temperature, high-salt, drought, and ABA conditions, but it was more susceptible to the effects of cold and drought. The overexpression of *MbWRKY53* enhanced the adaptability of *Arabidopsis* to cold and dry environments and can activate the expression of related genes through the CBF pathway and ABA-dependent pathway, thereby improving plant tolerance to these two stresses.

## Figures and Tables

**Figure 1 ijms-25-07626-f001:**
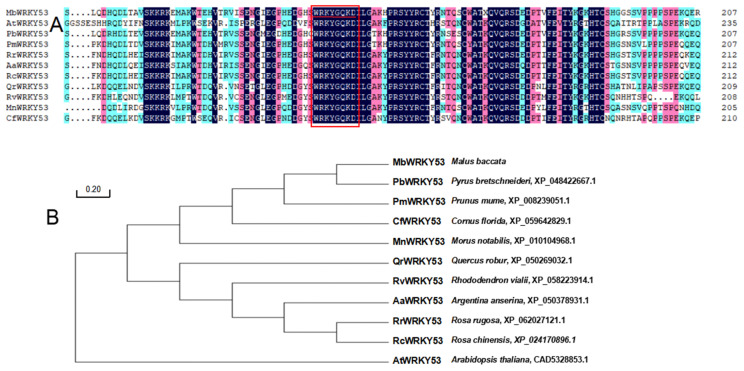
Analysis results of amino acid sequences for MbWRKY53 and other WRKY proteins (**A**), as well as phylogenetic tree analysis (**B**). Note: The target protein is marked with a red underline, and the WRKY domain is represented by a red box.

**Figure 2 ijms-25-07626-f002:**
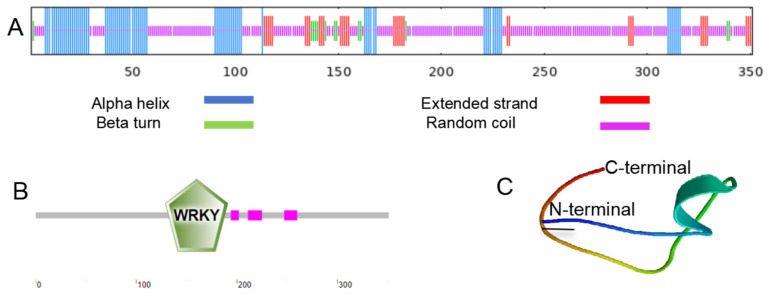
Prediction of the structure of the MbWRKY53 protein. (**A**) MbWRKY53′s structural analysis at the secondary level. (**B**) Conserved structural domain study of MbWRKY53. (**C**) Tertiary structural prediction of MbWRKY53. Predicting protein secondary structure (1 July 2024), domains (1 July 2024), and tertiary structure (1 July 2024).

**Figure 3 ijms-25-07626-f003:**
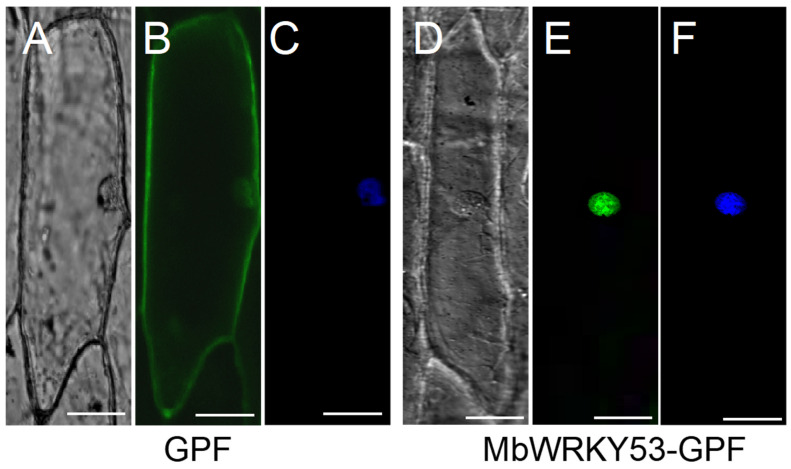
Subcellular localization of MbWRKY53. (**A**,**D**) Bright-field images, (**B**,**E**) GFP fluorescence, (**C**,**F**) the effects after DAPI dyeing. The scale in the figure represents 50 cm (Bar = 50 μm).

**Figure 4 ijms-25-07626-f004:**
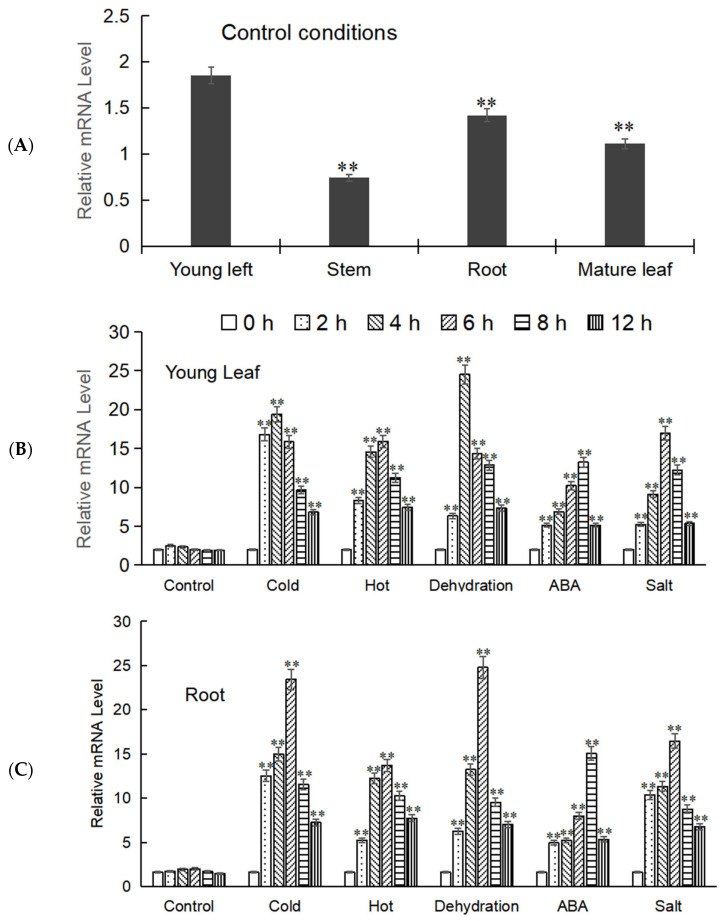
Expression analysis of *MbWRKY53*. (**A**) Expression of *MbWRKY53* in roots, stems, and leaves of *M. baccata*; (**B**,**C**) The expression level of *MbWRKY53* in young leaf (**B**) and root (**C**) under control conditions, cold, hot, deficient, ABA, and salt stress treatments with different time. The control indicates that no stress conditions were applied. The asterisks above the columns indicate significant differences and extremely significant differences compared to the control group (CK) (** *p* ≤ 0.01).

**Figure 5 ijms-25-07626-f005:**
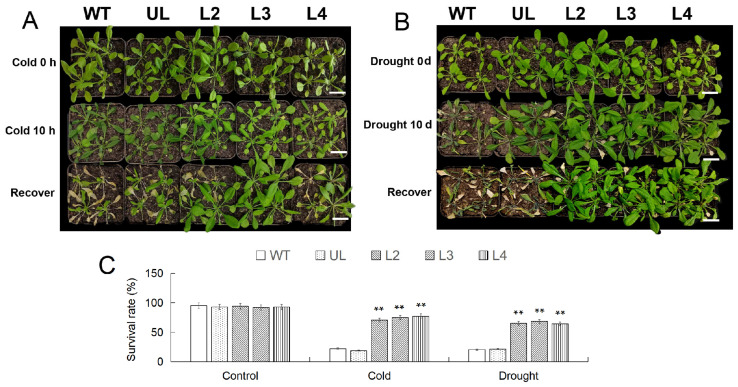
The phenotypes of *Arabidopsis* before and after 10 h of low-temperature treatment and after 7 d of recovery at room temperature (**A**); the phenotype of *Arabidopsis* after 10 d of drought treatment and 7 d of recovery under normal conditions (**B**); changes in survival rate of *Arabidopsis* (**C**). The length of the scale in the figure represents 3 cm (Bar = 3 cm). ANOVA compared the difference in survival rate between CK and transgenic lines (** *p* ≤ 0.01); the survival rate of WT was used as a control.

**Figure 6 ijms-25-07626-f006:**
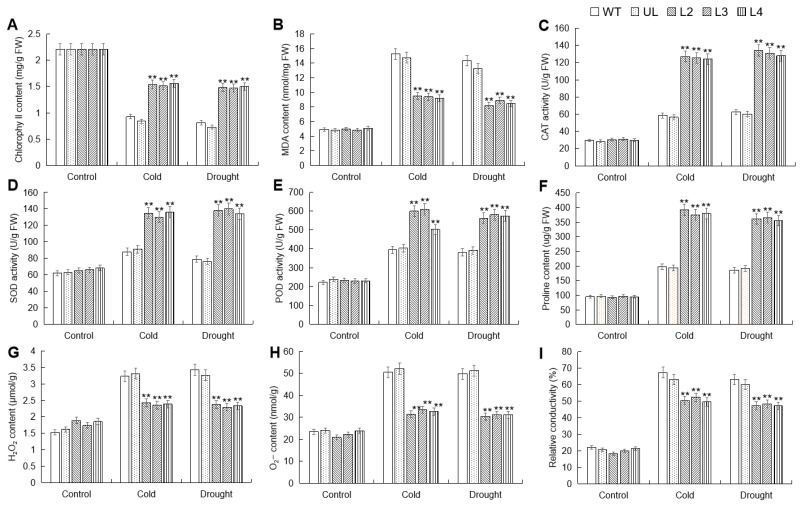
Changes in physiological indicators of WT, UL, and *MbWRKY53*-OE lines before and after treatment with cold and drought stress: (**A**) chlorophyll content; (**B**) MDA content; (**C**) CAT activity; (**D**) SOD activity; (**E**) POD activity; (**F**) proline content; (**G**) H_2_O_2_ content; (**H**) O_2_^−^ content and (**I**) relative conductivity. All data are the average of 3 repetitions; ANOVA compared the differences in relevant physiological indicators among the transgenic lines, UL, and WT (** *p* ≤ 0.01), which are parameters in WT used as controls.

**Figure 7 ijms-25-07626-f007:**
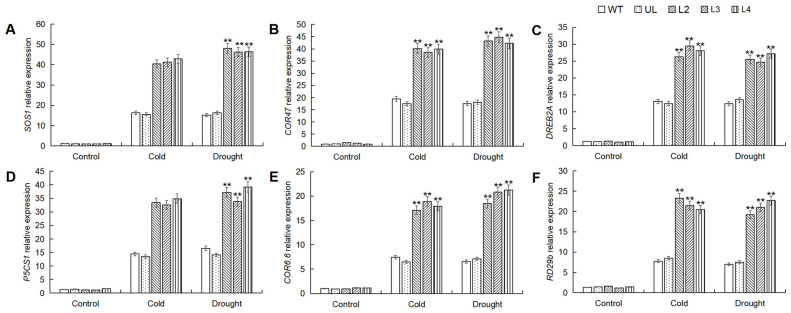
Analysis of the expression levels of target genes related to cold and drought in the WT, UL, and *MbWRKY53*-OE *Arabidopsis* lines before and after treatment with cold and drought: (**A**) *SOS1*; (**B**) *COR47*; (**C**) *DREB2A*; (**D**) *P5CS1*; (**E**) *COR6.6*, and (**F**) *PD29b*. Data represent means and standard errors of three replicates; ANOVA compared the differences in gene expression levels among the transgenic lines, UL, and WT (** *p* ≤ 0.01); gene expression levels in WT were used as controls.

**Figure 8 ijms-25-07626-f008:**
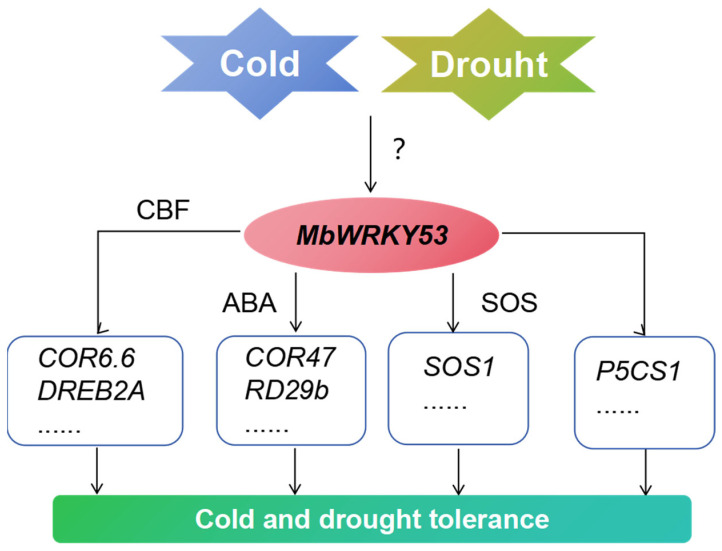
Potential pathways of *MbWRKY53* gene regulation in plant response to low temperatures and drought stress.

## Data Availability

The original data for this present study are available from the corresponding authors.

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
