# Peer review of "MbWRKY53, a M. baccata WRKY Transcription Factor, Contributes to Cold and Drought Stress Tolerance in Transgenic Arabidopsis thaliana"

_ijms, 2024, doi:10.3390/ijms25147626_

Round 1

Reviewer 1 Report

Comments and Suggestions for Authors

Authors studied the impact of  MbWRKY53 trans-factor on cold and drought tolerance of transgenic Arabidopsis. Initial part of study included the cloning of cDNA for MbWRKY53, its domain structure, pkylogenetic analysis, conserved motif distribution and in silico protein structure modeling. Authors tested the expression of  MbWRKY53 in different organs of Malus baccata (steam, roots, young and old leaves) after exposition to stress conditions (cold, heat, dehydration, salt and abscisic acid) by qRT-PCR. The nuclear localization of cloned trans-factor was confirmed by confocal fluorescence microscopy.

Then Authors tested the response of Arabidopsis plants (three lines) overexpressing MbWRKY53 to the cold and drought stress.

Authors analysed also the concentration or activity of malonyldialdehyde, proline, catalase, superoxide dismutase and peroxidase to explain the observed resistance.

Finally Authors tested the expression of several genes associated with cold and drought resistance in lines overexpressing MbWRKY53.

In my opinion artcicle is well written and the methodology is appropriately selected. Results support conclusions. The relatively weaker part is the lack of the direct proof of MbWRKY53 influence on selected drought- and cold-responsive genes for ex ample by Y1H or EMSA assay. However, adding the results of in silico analysis- as for example the presence of W-box or related cis-active motif in promoters of analysed genes (SOS1, COR47, DREB2A, P5CS1, COR6.6, and RD29b) could support obtained results.   

Detailed points that should be improved are as follows:

1. Fig 2A and 6F

Remove Chinese signs from the picture.

2. Fig 2B

Describe what are the domains other than WRKY?

3. Fig 2C

Provide localization of N- and C- terminal end of protein.

4. Fig 3- provide the size of scale bar

5. Line 180-181; do not capitalize malonyldialdehyde, catalase and peroxidase.

6. Line 277. Abbreviation: chlorophyl (chl) should be included.

7. Section 2.7 Authors could try to find if promoters of genes tested in section 2.7 (SOS1, COR47, DREB2A, P5CS1, COR6.6, and RD29b) contain W-box or related cis-elements. It is no 100% sure proof that tested trans-factor or related WRKY regulates their expresion, but make presented conclusions more probable to occur.

8. Section 4.5 Provide the emission and excitation filters range (in nanometers) 4.used in the study

9. Section 4.6 Provide the expected length of PCR products for tested and reference gene.

Provide citation of previous use of actin gene as a refernce or analysis of stability expression Rusing BestKeeper or related software. Provide information from which species originates the actin gene. Do not capitalize actin.

10. Section 4.7

Should be KpnI not kpnI.

Agrobacterium tumefaciens write in italics.

11. Line 423 write proline or Pro.

12. Section 4.9

How the purity of RNA was assessed?

How the remnants of genomic DNA were removed?

Provide approximate Mount of RNA/cDNA for one qPCR sample

Provide details of PCR reacton

Name of PCR equipment- manufacturer, country of origin.

Software used to acquire the raw qPCR data

Method used to analyse qPCR results with citation, for example Livak and Schmittgen or other.

13. Other comments:

Lines 113-114 double use of consitent, exchange one word consistent with synonym as concordant or coherent or consonant.

Rewrite the following sentence in lines 324-325

After two weeks, select seedlings with good growth and place them in Hoagland nutrient solution for hydroponic treatment.

To following or related:

After two weeks, seedlings with good growth rate were selected and placed in Hoagland nutrient solution for hydroponic treatment.

Comments on the Quality of English Language

Minor editing of English language required.

Reviewer 2 Report

Comments and Suggestions for Authors

There is a bit of a problem with the title, so please specify it as Arabidopsis thaliana.

Fig 1: The authors indicated the WRKY domain, but it would be better to specify only the 'WRKY' motif. Also, please create a phylogenetic tree with the highest possible resolution.

Fig 2: Using Chinese is problematic for an international journal, so please adhere to the standard format. Additionally, panel A contains data that can be combined with panels B and C, and it is unnecessary to present it as the main data.

Fig 3: There is a significant error in that it shows 35S

. It should be corrected to GFP. This seems to be a major oversight by the authors in data preparation.

Fig 4: An explanation is needed regarding the significant expression for each tissue. Additionally, please clarify what is meant by 'control conditions'.

Fig 5: I have initiated an experiment using transformants, but information about their expression is needed. If possible, please include an experiment using transformants that have been 'tagged' and analyzed at the protein level.

Fig 6: This figure also contains Chinese, which needs correction.

The schematic diagram in Fig 8 represents the authors' creative idea. However, using a simple box-type flowchart reduces the quality of the paper. I recommend using a more sophisticated illustration.

Overall, this paper focuses on the characterization of the apple WRKY transcriptional regulator, but it seems quite far from the standards of this journal. The authors should also pay more attention to the quality of the data, which is currently insufficient.

Round 2

Reviewer 1 Report

Comments and Suggestions for Authors

Authors strongly improved the manuscript according to suggestions and replied to all questions.

Only minor corrections should be introduced:

1. Provide the description of Fig. S1 and S2, put it under these Figures.  

2. On the axis 0Y in Fig. S2 provide the name of gene, i.e. put the information: relative expression of….. gene

3. Provide sequences of primers for reference Mbactin gene in table S1.

4. Provide the length of PCR roducts for tested and reference gene in table S1.

4. In line 282/283 use only abbreviation chl not the full name chlorophyl. Remove the name chlorophyl. The abbreviation was introduced earlier in line 184, there is no need to do it for the second time.

Comments on the Quality of English Language

Minor editing of English language required.
